# A Study of the Power of Heuristic-based Pruning via SAT Planning

**Christopher Johnson, Pascal Bercher, Charles Gretton**

Australian National University
{christopher.johnson, pascal.bercher, charles.gretton}@anu.edu.au

## Abstract

Planning as SAT (satisfiability) is the method of representing a horizon-bounded planning problem as a Boolean SAT problem, and using a SAT decision procedure to solve that problem. Representations are direct, thus a solution plan can be obtained directly from a satisfying valuation. By querying a SAT solver over a series of horizon lengths, up to a completeness threshold, this approach can be the basis of a complete planning procedure. SAT planning algorithms have been theoretically contrasted with IDA$^*$ search, a heuristic state-based search algorithm, where a theoretical exponential separation is demonstrated in favour of the SAT approach. Here a nominated heuristic is implemented in SAT with the query formulae encoding heuristic information.

We make two practical contributions related to this background. First, we provide to the best of our knowledge the first practical implementation of a theoretical SAT encoding of the $h^2$ heuristic. Second, we empirically evaluate SAT-based pruning by implementing heuristics $h_{max}$ and $h^2$.

## Introduction

The research topic of encoding planning tasks in SAT is well understood, and numerous methods have been proposed for representing the classical deterministic planning problem in SAT (Kautz and Selman 1992, 1996; Kautz, McAllester, and Selman 1996; Ernst, Millstein, and Weld 1997; Biere et al. 1999; Rintanen, Heljanko, and Niemelä 2006; Robinson et al. 2009; Höller and Behnke 2022). Encoding planning search *heuristics* in SAT formulae has proven to be less straightforward, and is a topic that has received relatively little attention in the literature. If you *implement* a heuristic in the SAT representation of the horizon-bound problem, a fast procedure called *unit propagation* (UP) is able to efficiently deduce that the goal cannot be achieved. Rintanen (2011) proves that if a heuristic has been implemented in the SAT formula encoding of a planning task, solving the problem via the standard SAT workflow can simulate Iterative Deepening A* (IDA*) (Korf 1985) within the same time and space bounds. Rintanen also shows that it is possible to construct a planning problem such that A* and IDA* take exponentially more computation to find a solution than the aforementioned SAT-based approach using a UP procedure to prune the set of explored models.

It is possible to combine a state-based forward search, using a bounded search algorithm such as IDA*, and the UP-powered heuristic-based pruning. Each time a state is discovered, we can perform UP on the combined SAT formula of the planning task, heuristic, and encoding of the search state, and if this formula is found to be unsatisfiable, one can completely prune the state from the search – i.e., because we know a goal cannot be reached from that state given the horizon bound. With parts of the search space thereby pruned away, we can potentially improve the efficiency of finding plans contrasting to a regular state-space search without such pruning. Whether or not this method results in an overall advantage—or whether the overhead of implementing and running such processes dominates any resulting efficiency gains—is an interesting question that we investigate in this work. Our empirical work thus far is faithful to the analytical results established by Rintanen (2011), thus here we do not explore the many possibilities of synergies with search optimisation techniques and advanced approaches to representing problems and knoweldge about problems.

## Definitions and Notations

We use the SAS$^+$ (Sandewall and Rönnquist 1986) planning formalism, in which variables that characterise state are multi-valued, being assigned one value from a finite domain, instead of being restricted to the binary domain of *True* and *False* as in STRIPS (Fikes and Nilsson 1971). We assume a uniform action cost setting, where all actions have cost 1 – i.e., the cost of a plan is its serial length. The following definition of a SAS$^+$ planning task is adapted from Huang, Chen, and Zhang (2012).

**Definition 1 (SAS$^+$ Planning)** *An instance of a planning task* $\Pi$ *in SAS$^+$ is a tuple* $\Pi = \langle X, A, s_0, G \rangle$, *where:*

- $X = \{x_1, \ldots, x_n\}$ *is a set of multi-valued variables, each with a finite domain* $\mathrm{dom}(x)$. *A variable assignment is written* $x \mapsto v$ *where value* $v \in \mathrm{dom}(x)$. *We also refer to such a variable assignment as a **fact**, forming the set of facts* $F = \{x \mapsto v \mid x \in X, \ v \in \mathrm{dom}(x)\}$.

- $A$ *is a set of actions. Each action* $a \in A$ *has two associated sets of partial variable assignments* $\mathrm{pre}(a)$ *and* $\mathrm{eff}(a)$, *representing action preconditions and effects respectively.*

- *A state* $s$ *is a full assignment (a set of assignments that assigns a value to every state variable). If an assignment*

$x \mapsto v$ is in $s$, we can write $s(x) = v$. The set of all states is represented by $S$.

- $s_0 \in S$ is the initial state of the problem.
- $G$ is a partial assignment of variables that make up the goal conditions of the problem. A state $s \in S$ is a goal state iff $G \subseteq s$.

**Definition 2 (Action Application)** *An action $a$ is applicable in a state $s$ iff* $\mathrm{pre}(a) \subseteq s$, *i.e. the preconditions of the action are met. The successor state $s' = \mathrm{succ}_a(s)$ is the result of applying the action $a$ from state $s$, given it is applicable. The state $s'$ contains the same variable assignments as $s$ except for the corresponding assignments specified in $\mathrm{eff}(a)$, which replace the values in $s$. Formally:*

$$s' = \{(x' \mapsto v') \mid (x' \mapsto v') \in s \text{ s.t. } \forall (x \mapsto v) \in \mathrm{eff}(a) \; x \neq x'\} \cup \{(x \mapsto v) \in \mathrm{eff}(a)\}$$

*The successor function $\mathrm{succ}_a(s)$ is not defined when $s$ does not satisfy the preconditions of $a$, making it a partial function. This means that actions that are not applicable to a state cannot be used from that state.*

**Definition 3 (SAS$^+$ Delete Set)** *We define $\mathrm{del}(a)$ as the SAS$^+$ analogue of the delete set used in STRIPS, which represents all the facts that are no longer true after the action $a$ is applied. Formally we can define this as:*

$$\mathrm{del}(a) = \{(x \mapsto v) \mid (x \mapsto v) \in F, (x \mapsto r) \in \mathrm{eff}(a), r \neq v\}$$

*In other words, for each effect fact in which a variable $x$ is assigned to value $v$, $\mathrm{del}(a)$ includes all other facts that assign the variable $x$ to a value other than $v$.*

**Definition 4 (Sequential Plan)** *A sequential plan $\pi$ of length $n \in \mathbb{N}$ for a planning task $\Pi$ is a sequence of actions $\pi = [a_1, \ldots, a_n]$ for which $G \subseteq \mathrm{succ}_{a_n}(\mathrm{succ}_{a_{n-1}}(\ldots(\mathrm{succ}_{a_1}(s_0))))$. We explicitly ignore the degenerate case of the empty plan, resulting from $G \subseteq s_0$. $|\pi|$ denotes the length of the plan $\pi$. A sequential plan is* **optimal** *if there does not exist any shorter plan $\pi'$ for which $|\pi'| < |\pi|$.*

## Encoding SAS+ in Boolean SAT

We assume familiarity with (Boolean) SAT solving and related concepts in propositional logic. In order to leverage SAT solving to help find plans in a planning task, we need a way to encode the planning task as a CNF Boolean formula. Representing a planning task in propositional logic was first done by Kautz and Selman (1992), with various other representations created since, with the example by Huang, Chen, and Zhang (2012) somewhat pertinent here. We will be using a version of the Direct Encoding (Balyo 2014), which uses a unique propositional variable for each action and variable assignment at each discrete timestep. If one of these propositions is true, this means that the action or variable assignment holds (is true at a state) at that timestep, and does not hold (is false at a state) if it is false. The direct encoding also dictates how these propositions can change between two consecutive timesteps. It is necessary for the planning task to have a bound on plan length (the horizon) in order for the encoding to have a finite number of variables, as each

fact and action needs its own propositional variable for every possible timestep of the planning task.

Specifically, we want to create a CNF formula $H_T$ such that $H_T$ is satisfiable iff there exists a valid plan of length at most $T$ in our original planning task $\Pi = \langle X, A, s_0, G \rangle$. Additionally, our selected encoding is constructive and therefore relatively simple to extract the plan if a satisfying assignment for $H_T$ is found. The encoding uses two kinds of Boolean variables:

- *Action variables* $a@t$, which indicate whether the action $a$ is used at the $t$-th timestep, for each action $a \in A$ and each timestep (not including the horizon itself) $t \in \{0, \ldots, T-1\}$.
- *Fact variables* $f_{x \mapsto v}@t$, which indicate whether variable $x$ is assigned the value $v$ at the start of timestep $t$ (before the actions of the timestep are applied), for each variable $x \in X$, value $v \in \mathrm{dom}(x)$, and timestep (including the horizon) $t \in \{0, \ldots, T\}$.

Using these variables, we can now provide the clauses that $H_T$ is comprised of. Each state variable has a value:

$$(f_{x \mapsto v_1}@t \vee f_{x \mapsto v_2}@t \vee \cdots \vee f_{x \mapsto v_d}@t)$$
$$\forall x \in X, \; \mathrm{dom}(x) = \{v_1, v_2, \ldots, v_d\}, \; \forall t \in \{0, \ldots, T\}$$

Each state variable has at most one value:

$$(\neg f_{x \mapsto v_i}@t \vee \neg f_{x \mapsto v_j}@t)$$
$$\forall x \in X, \; v_i \neq v_j, \; \{v_i, v_j\} \subseteq \mathrm{dom}(x), \; \forall t \in \{0, \ldots, T\}$$

If an action is taken at a step $t$, then its preconditions must hold at the beginning of step $t$:

$$(\neg a@t \vee f_{x \mapsto v}@t)$$
$$\forall a \in A, \; \forall (x \mapsto v) \in \mathrm{pre}(a), \; \forall t \in \{0, \ldots, T-1\}$$

If an action is taken at step $t$, then its effects must hold at the start of step $t+1$:

$$(\neg a@t \vee f_{x \mapsto v}@t+1)$$
$$\forall a \in A, \; \forall (x \mapsto v) \in \mathrm{eff}(a), \; \forall t \in \{0, \ldots, T-1\}$$

Variable assignments do not change between two consecutive timesteps *unless* some action has changed them (known as frame axioms):

$$(f_{x \mapsto v}@t \vee \neg f_{x \mapsto v}@t+1 \vee a_{s_1}@t \vee \cdots \vee a_{s_j}@t)$$
$$\forall x \in X, \; v \in \mathrm{dom}(x), \; \mathrm{support}(x \mapsto v) = \{a_{s_1}, \ldots, a_{s_j}\},$$
$$\forall t \in \{0, \ldots, T-1\}$$

Here, $\mathrm{support}(x \mapsto v) \subseteq A$ is the set of *supporting actions* of the assignment $x \mapsto v$. An action $a$ is a supporting action of an assignment $x \mapsto v$ if the assignment is one of the effects of the action, i.e. $(x \mapsto v) \in \mathrm{eff}(a)$. Because we are looking for a sequential plan, we must forbid any two actions from being executed at the same timestep:

$$(\neg a_i@t \vee \neg a_j@t)$$
$$\forall \{a_i, a_j\} \in A, \; a_i \neq a_j, \; \forall t \in \{0, \ldots, T-1\}$$

Finally, the initial state assignments must be true at the first timestep, and the partial goal state assignments at the final timestep:

$$\bigwedge_{x \mapsto v \in s_0} f_{x \mapsto v}@0 \wedge \bigwedge_{x \mapsto v \in G} f_{x \mapsto v}@T$$

The final formula $H_T$ is the conjunction of each set of formulae described above. If $\alpha$ is a satisfying assignment of $H_T$, then a valid plan can be extracted directly from $\alpha$, as the $t$-th action of the plan sequence is the action $a \in A$ of which $\alpha(a@t) = \textit{True}$. In other words, the action that is true at each timestep forms the sequence of actions in the plan. If the plan includes no-ops, then there will exist at least one timestep $t$ in which $a@t = \textit{False}$ for all actions, representing a timestep where a no-op action is taken. For the proof that this encoding is satisfiable if and only if the original planning task has a valid plan, see Balyo (2014).

We define $H_{T-G}$ as $H_T$ without the final clause set describing the goal conditions. We use $S@t$, where $S$ is any set of facts, to mean the conjunction $f_1@t \wedge \ldots \wedge f_n@t$ where $\{f_1, \ldots, f_n\} = S$. Using these definitions, we can define the optimal planning problem as finding a $T$ such that $H_{T-G} \wedge G@T$ is satisfiable but $H_{T-G} \wedge G@T-1$ is not. We can also define the satisficing planning problem (in which any plan is to be found, not necessarily the best one) as simply whether or not a $T$ exists that satisfies $H_T$. These definitions will allow us to make more general statements about the SAT encoding in the next section.

## Implementing Heuristics in SAT

Heuristics are the defining factor for the efficiency of informed search algorithms. A good heuristic provides additional information on the goal conditions that can guide the search towards a plan faster, while also being able to preemptively rule out entire branches of the search if it is known that a goal cannot be found within a given horizon. So far we have defined an encoding of a planning task as SAT, but currently it does not obviously contain any heuristic information. In this section, we begin by defining what it means to "implement" a heuristic in a SAT encoding, before exploring two such heuristic implementations.

At the heart of what we will write about is Unit Propagation (UP), an efficient inference procedure that plays an important role in systematic SAT solving algorithms. Applying UP exhaustively on a formula can be done in linear time proportional to the number of clauses in the formula. This efficiency makes it a prime candidate for how we define the implementation of heuristics in a SAT solving procedure.

The aim for implementing a heuristic in SAT is to find a way to identify states where the heuristic lower bound signifies that reaching a goal from that state is impossible to do before hitting the horizon first.

If performing UP on a set of clauses $C$ results in a clause $c$, we denote this by $C \vdash_{UP} c$. The *implementation* of an admissible heuristic in a SAT encoding of a planning task is defined as follows (Rintanen 2011):

**Definition 5 (Implemented Heuristic)** *Let $f$ be a fact and $T$ the horizon bound. A clause set $\chi_T$ implements the admissible heuristic $h^f(s)$ if for all $t \in \{0, \ldots, T\}$, all states $s$, and all $t' \in \{t, \ldots, min(T, t + h^f(s) - 1)\}$, we have $s@t \wedge H_T \wedge \chi_T \vdash_{UP} \neg f@t'$.*

For more clarity, this definition can be generalised for any set of goals $G$ as follows (Huang 2012):

**Definition 6 (Generalised Implemented Heuristic)** *A set of clauses $\chi_T$ implements the heuristic function $h^G(s)$, for a given horizon $T$, if $s@t \wedge H_T \wedge \chi_T \vdash_{UP} \bot$ for all states $s$ and times $t \in \{0, \ldots, T\}$ such that $(T - t) < h^G(s)$.*

In other words, an admissible heuristic is implemented for a given horizon if for all states, UP will derive an empty clause for each timestep in which the distance between that timestep and horizon is less than the heuristic value of achieving a goal from that state.

## Encodings that Implement Heuristics

The $h_{max}$ heuristic (Bonet and Geffner 2001), generally speaking, is concerned with the cost of the most expensive subgoal. Interestingly, creating the $\chi_T$ formula for the $h_{max}$ heuristic implementation is unnecessary, because we actually get this for free in the problem encoding $H_T$ itself. See (Rintanen 2011) for the full proof of this property. Formally, we can say that the empty clause set $\chi_T = \emptyset$ for any $T \geq 0$ implements $h^f_{max}$ for any fact $f$.

The $h^m$ set of heuristics (Haslum and Geffner 2000) are similar to $h_{max}$ in the sense that they are also estimating the distance to the entire set of goal facts via the cost of achieving smaller subsets of facts. Whereas $h_{max}$ is concerned with the single most expensive fact, $h^m$ uses the cost of achieving the most expensive subset of facts of size at most $m$. In this sense, we can view $h_{max}$ as being a special case of $h^m$, where $h_{max} = h^1$.

**Definition 7 ($h^m$)** *The $h^m$ heuristic value for achieving the set of facts $\psi \in F$ from state $s$ is:*

$$h^m_\psi(s) = \begin{cases} 0 & \textit{if } \psi \subseteq s \\ min_a h^m_{\mathrm{R}(\psi,a)}(s) + 1 & \textit{if } |\psi| \leq m \\ max_{\psi' \subsetneq \psi, |\psi'|=m} h^m_{\psi'}(s) & \textit{if } |\psi| > m \end{cases} \quad (1)$$

*where $\mathrm{R}(\psi, a)$ is the **regression operator**, defined as $\psi \setminus \mathrm{eff}(a)) \cup \mathrm{pre}(a)$ if both $\mathrm{eff}(a) \cap \psi \neq \emptyset$ and $\mathrm{del}(a) \cap \psi = \emptyset$, and is undefined otherwise.*

In this definition, the min ranges over all actions $a$ such that $\mathrm{R}(\psi, a)$ is defined. Also recall from our SAS$^+$ definition that the $\mathrm{del}(a)$ set contains the facts that are no longer true once the action is applied. To obtain the $h^m$ heuristic value for the current state to a goal state, simply take the heuristic value on the set of goal facts $h^m_G(s)$.

The $h^m$ heuristic is unfortunately not implemented for free like $h_{max}$ is, therefore we must create a non-empty clause set $\chi_T$ to do so. The implementation we outline here was described by Huang (2012), and is defined as the conjunction of multiple clause types, similar to the Direct Encoding. We label these clauses as Types 1 through 4.

In order to implement $h^m$, we need Boolean variables that correspond to the truth of a set of facts. Specifically, for every set of facts $\phi \subseteq F$ such that $2 \leq |\phi| \leq m$, we create a *meta-fact* $f_\phi$, which implies the truth of all facts in $\phi$. We begin outlining the clauses of $\chi^m_T$ by encoding this property in SAT:

$$(\neg f_\phi@t \vee f@t)$$
$$\forall t \in \{0, \ldots, T\}, \forall f \in \phi \quad \text{(Type 1)}$$

In the case when $\phi = \{f\}$, $f_\phi$ is just an alias for $f$, thus $f_\phi$ is defined for all $\phi \subseteq F$ such that $1 \le |\phi| \le m$.

Next we define the frame axiom for each meta-fact, which gives the condition under which the value of $f_\phi$ can change from *False* to *True*. It consists of the following clauses for each $t \in \{0, \ldots, T{-}1\}$:

$$(f_\phi @ t \vee \neg f_\phi @ t{+}1 \vee x_1 \vee \ldots \vee x_p) \qquad \text{(Type 2)}$$

$$\begin{array}{c}(\neg x_i @ t \vee f_{\phi'} @ t) \\ \forall \phi' \subseteq \mathrm{R}(\phi.a_i) \neq \emptyset, \\ |\phi'| = \min(m, |\mathrm{R}(\phi, a_i)|), \forall i \in \{1, \ldots, p\} \end{array} \qquad \text{(Type 3)}$$

where $a_1, \ldots, a_p$ are all the actions for which $\mathrm{R}(\phi, a_i)$ is defined, and $x_1, \ldots, x_p$ are a set of auxiliary variables. In other words, clause set Type 3 represents that if a set of facts has changed from *False* to *True*, this means that for at least one action that applies to that set of facts, a subset of facts (of size at most $m$) obtained from regressing that action must be *True*. It is perhaps worth noting that the auxiliary variable may not necessarily be distinct, as $x_i$ and $x_j$ are the same variable if $\mathrm{R}(\phi, a_i) = \mathrm{R}(\phi, a_j)$. Also whenever $\mathrm{R}(\phi, a_i) = \emptyset$, the set of clauses Type 3 is empty but $x_i$ appears in clause Type 2 regardless. This represents cases where the set of facts $\phi$ can be achieved by an action with no preconditions.

The last clause set is to ensure that the goals are met in terms of meta-facts:

$$\begin{array}{c}(f_{\phi'} @ T) \\ \forall \phi \subseteq G, 2 \le |\phi| \le m \end{array} \qquad \text{(Type 4)}$$

Together, the conjunction of clause sets Type 1, Type 2, Type 3, and Type 4 define $\chi_T^m$, the implementation of the $h^m$ heuristic in SAT. For the proof of this, see Huang (2012).

## Search with SAT-based Heuristic Pruning

We describe a heuristic forward search of the problem state space that will make use of an implementation of a heuristic in SAT to prune states from the search. We begin at the point where the search software has parsed the planning task, translating the problem to a $\mathrm{SAS}^+$ representation and preparing to begin the state space forward search. We have seen that a heuristic can be implemented in SAT so that inference by UP is able to determine when the state cannot reach the goal given a horizon bound. Picture the case where we happen to be in the middle of a bounded forward search and decide to pass the current state of the search to a SAT solver as described before. If UNSAT is obtained from the given clause set, then we can effectively prune this state from the search. In Figure 1 we provide a graphical example of how such a pruning step can occur.

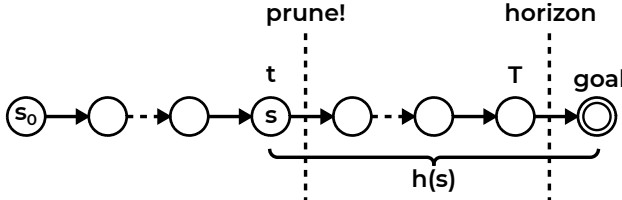

Figure 1: For the state $s$, $(T - t) < h(s)$, and so $s@t \wedge H_T \wedge \chi_T \vdash_{UP} \bot$ (provided that $h$ is an admissible heuristic). This UNSAT result tells us that the state cannot reach any goal within the horizon, so we can prune the search at that state.

For an implementation of this idea, before the search is initiated we extract the variables (and their domains), actions (and their preconditions and effects), goal conditions, and the initial state. For every timestep up until the horizon, every fact (i.e. every possible value assignment for each variable) is associated with a unique Boolean variable. Additionally, for every timestep up until but not including the horizon, each action is associated with a unique Boolean variable, distinct from the set of fact variables. These represent the Boolean variables introduced in the Direct Encoding. Using these variables, all of the clauses defined by the formulae of our SAT encoding for $\mathrm{SAS}^+$—hereby referred to as the *base encoding*—are generated and passed to the SAT solver. These clauses will never change during the search, as they represent the ground truth information described by the problem domain and instance, i.e. they hold irrespective of the current search state. The base encoding represents the clause set $H_T$ described earlier.

Next, we generate the *heuristic encoding* $\chi_T$ using the same set of fact and action variables. The clauses generated in this step are added to the set of clauses from the base encoding, and likewise do not change throughout the search process. Our approach initialises an incremental SAT solver with a formula of the form $H_T \wedge \chi_T$, encoding the base planning problem $H_T$ along with the implemented heuristic $\chi_T$. We assume familiarity with incremental SAT (Fazekas, Biere, and Scholl 2019). Notice at this point that we have two heuristic "implementations" at work here: *(i)* the *search heuristic*, which evaluates each state with a heuristic value in order to prioritise which states in the search frontier should be explored first, and *(ii)* what we call the *prune heuristic*, which is what is encoded in the SAT formula $\chi_T$ in order to prune states that cannot reach a goal state before hitting the horizon according to the heuristic value of the state. The heuristic used for state prioritisation in search, and that used for pruning, do not need to be the same. Any heuristic can be used for either, provided that the prune heuristic is admissible and there exists a SAT encoding of it that satisfies the definition of an implemented heuristic.

The SAT-based pruning step occurs when the search considers adding a new state of the frontier. The incremental SAT solver is queried for every such state, with the query assumption corresponding to the unit clauses asserting $s@t$. Because a state $s$ is defined as a set of facts $\{f_1, f_2, \ldots, f_n\}$ that are true at that point in time, and due to the $\mathrm{SAS}^+$ definition requiring every variable in a state to be set to a particular

value, we can take the literal of each fact in the set and create a conjunction of unit clauses $f_1@t \land f_2@t \land \ldots \land f_n@t$, forcing each of those facts to be true in the combined SAT formula. The value of $t$ in this case (representing the timestep of the plan in which these facts are true) is set to the $g(s)$ value – i.e., $g(s)$ is the timestep that the state has been discovered at since $s_0$. With all of this in place, the incremental SAT solver is modified to return three possible results:

1. UNSAT: The solver has detected an empty clause during the UP procedure. There is no way to reach a goal before hitting the horizon from this state, due to the condition $s@t \land H_T \land \chi_T \vdash_{UP} \perp$ (from the generalised definition) being satisfied. In this case, the state is not added to the frontier, effectively pruning it from the search, and the search continues.

2. SAT: The solver has found an assignment for every literal that satisfies the formula (and thus $s@t \land H_T \land \chi_T \nvdash_{UP} \perp$). This means that the SAT solver has managed to find a valid plan for the original problem during the UP procedure. In this case, because the planning task is now solved and there is no need to continue, the entire planning process can be terminated. All that is remaining is to extract the valid plan from the literal assignments, which can be done as follows: for every literal corresponding to an action variable $a@t$, if its assignment is *True*, then action $a$ is performed at timestep $t$ in the plan. Due to how $H_T$ is defined, there will be exactly one action for each timestep $t \in \{0, \ldots, T-1\}$, or in cases where no-op actions are included, *at most* one action.

3. UNKNOWN: If neither UNSAT nor SAT are returned then the result is unknown, as the solver did not solve the formula nor did it generate any empty clauses within the conflict budget – i.e., using UP only. This means that UP via the implemented heuristic cannot detect that it is certain the horizon will be reached before any goal. It is still entirely possible that a goal state is impossible to reach from this state, due to the horizon or otherwise, but at the very least the heuristic information is not sufficient to deduce this with UP. In this case, the intercepted state is added to the frontier as usual, and the search continues.

If the planning task remains unsolved after the SAT query, the search continues as normal: a state is popped off the frontier queue based on the search heuristic priority, and unexplored states adjacent to it are ready to be added to the frontier, being intercepted by our pruning process before they can do so. This continues for every explored state until either the search finds a goal state, the SAT solver returns SAT, or the entire state space is explored (and therefore there does not exist a valid plan for the planning task at hand). This entire process of search with heuristic-based pruning can be seen in Algorithm 1.

This approach begs the question of how exactly such a procedure can be applicable, as a single "bounded search" is rarely seen in practice when one can simply continue the search until a goal is eventually found. Aside from search under strict memory limitations, one particular use-case (and the use-case we are particularly interested in) is IDA*, which conducts a series of depth-bounded searches

```
1  create_literals();
2  H_T := generate_base_clauses();
3  χ_T := generate_heuristic_clauses();
4  s_0.depth := 0;
5  push s_0 onto queue;
6  mark s_0 as visited;
7  while queue not empty do
8  │   s := pop state from queue;
9  │   if s ⊇ G then
10 │   └   return get_plan(s)
       // -=+=- Begin Pruning -=+=-
11 │   s@t := generate_state_assumptions();
12 │   result := {H_T, χ_T}.unit_propagate(s@t);
13 │   if result == UNSAT then
14 │   └   continue;
15 │   if result == SAT then
16 │   │   plan := [];
17 │   │   for each t in {0..T−1} do
18 │   │   │   for each a in A do
19 │   │   │   │   if a@t then
20 │   │   │   │   │   plan.add(a@t);
21 │   │   │   │   └   break;
22 │   └   return plan;
       // -=+=-  End Pruning  -=+=-
23 │   if s'.depth < T then
24 │   │   for a in {a | pre(a) ⊆ s} do
25 │   │   │   s' := the result of applying a from s;
26 │   │   │   s'.parent := s;
27 │   │   │   s'.action := a;
28 │   │   │   s'.depth := s.depth+1;
29 │   │   │   if s' is not visited then
30 │   │   │   │   mark s' as visited;
31 │   │   │   └   push s' onto queue;

32 return Search failed: state space exhausted;
```

**Algorithm 1:** State Space Forward Search with Heuristic-based Pruning

by definition. By implementing this "*heuristic pruning*" procedure during IDA*, we retain the benefits of an iterative-deepening search (i.e. beneficial for memory-constrained environments), while aiming to improve the time efficiency of the search itself by reducing the size of the state space at each iteration.

It could seem strange at first that using the same heuristic for search and pruning results in fewer expanded states than using only the search heuristic, as it may seem that the states that get pruned from the search would not be visited by the search in the first place (even without the pruning). There are a few examples of where this gap tends to occur. First, the case where UP actually manages to solve the entire SAT formula, allowing the search iteration to end prematurely. The other explanation for this behaviour stems from the fact that the heuristic-based pruning requires performing UP on both the heuristic encoding and the encoding of the entire

problem itself. This means that the pruning procedure has access to more information than the search heuristic alone, and can potentially result in states being pruned for reasons that the search heuristic is not aware of. For example, if no goal states exist within the current search bound, regular search will exhaustively search the state space, whereas the heuristic-based pruning is likely to notice early in the search that no goal is achievable before the horizon and will prune away most of the state space. In particularly lucky cases this occurs at the initial state, allowing the search to skip to the next iteration immediately. Either way, this results in fewer states expanded when using heuristic-based pruning, despite the same heuristic being used for search in both cases.

## Experimental Results and Analysis

For each of our experiments, a modified version of the planning system Fast Downward and an integrated SAT solver MiniSAT are used to perform the heuristic-based state pruning. Each experiment runs a number of *jobs* concurrently, where each job is assigned to a domain from the selection of domains (listed below), and is given a time limit and a memory limit. A job continues until either all problems in the domain are solved, the time limit is met, or the memory limit is met. In any case, the job stops working on the current domain and starts working on another domain that has not yet been taken by another job. These jobs continue until all domains have been worked on, the experiment ends, and data is collected.

A domain being "worked on" refers to attempting to solve each problem in the domain in sequence in an iterative-deepening fashion. Specifically, the job calls Fast Downward with the appropriate search heuristic, prune heuristic, and search algorithm as parameters at an initial bound, and the job moves onto the next problem if a solution is found. If no solution is found and the (potentially pruned) search space has been searched exhaustively, the job increases the bound by one and calls Fast Downward again with this new bound value. This repeats with progressively increasing bounds until a solution is found, or until the time/memory limit is met. It is worth noting that all of the selected domains contain only problems that are solvable within a finite bound, so we do not have to handle the case where a job is stuck trying to find a solution to an unsolvable problem. Additionally, domains typically sort their problems from easiest to hardest to solve, but this is not a guarantee. For our experiments, we solve problems lexicographically, which tends to be easiest to hardest for most domains.

We make one particular modification to the iterative-deepening method at this stage, motivated by the efficiency loss of starting every search at a bound of 1, often wasting time searching within unreasonable bounds. To help mitigate this, we choose the initial bound strategically. Because admissible heuristics always give a lower bound of the actual distance to a goal—and therefore never overestimate the goal distance—we can take an admissible heuristic value of the initial state and use that as our initial bound for the iterative-deepening search. Better yet, we can take *multiple* admissible heuristics, obtain the value of the initial state from each, and take the maximum of those to get as close

as possible to the optimal distance (and bound). We achieve this by simply calling Fast Downward with a bound of 1 before each problem is solved, and extracting the heuristic information from that run to be used for the first search iteration.

When a job is finished with a problem, we take the resulting output and extract the time taken to find a solution to the problem, and the number of states expanded (i.e. all successors of the state have been generated) during the search. We can use the time data to evaluate efficiency and expanded states data to display any early backtracking caused by our implementation.

For each of our experiments, we specifically run 10 jobs concurrently, with each job being given a 30 minute time limit and an 8 GB memory limit, chosen to match typical official International Planning Competition (IPC) settings. The hardware specifications of the machine that the experiments were run on are as follows: Intel(R) Xeon(R) Gold 6252 CPU @ 2.10Ghz, 92 cores, 196.46 GB of memory, and Linux 4.15.0-156-generic #163-Ubuntu SMP as the OS.

A selection of 24 planning domains were tested in each of the experiments, covering a wide range of problems and difficulty levels. However, only 15 of these domains had at least one problem solved amongst all of the experiments. The list of these 15 domains are: *agricola-sat18-strips*, *blocks*, *depot*, *driverlog*, *floortile-sat14-strips*, *logistics00*, *miconic*, *philosophers*, *pipesworld-notankage*, *pipesworld-tankage*, *rovers*, *satellite*, *sokoban-sat11-strips*, *telegraph*, and *zenotravel*.

All of the domains used aside from *philosophers* and *telegraph* can be found at the University of Basel AI Group Github alongside the rest of the packaged Fast Downward benchmark domains. The *philosophers* and *telegraph* domains (Fabre et al. 2010), originally used for IPC4, both model deadlock detection problems that are generated automatically by translating from models originally defined in PROMELA (Process/Protocol Meta Language). Deadlock problem instances were generated with an increasing number of philosophers/stations, with the initial problem of each domain containing 2.

### Results and Evaluation

This section aims to display some relevant results obtained from the experiments and to evaluate the effectiveness of our approach. We run the experiments using the IDA* search algorithm, exploring the practicality of an existing theorem about SAT-implemented heuristics in the context of our heuristic-based pruning. The set of experiments detailed here are concerned with exploring the difference in solve time and expanded states among a collection of search heuristics, with and without heuristic pruning. We are also particularly interested in the specific case where the search heuristic and prune heuristic are the same, and how this compares to having no prune heuristic at all. This aims to test whether there exists a significant separation of A* and SAT in practice, similar to the exponential separation result in (Rintanen 2011) Theorem 10: "*state space search with A\* and a heuristic h is sometimes exponentially slower than any*

*algorithm for SAT that uses unit resolution, if the latter implements h".*

To make this section clearer in its distinction between discussion of search heuristics and prune heuristics, we will refer to the $h_{max}$ (i.e. $h^1$) prune heuristic as $\chi^1$, and the $h^2$ prune heuristic as $\chi^2$.

Specifically, we run IDA* with the $h_{max}$ and $h^2$ search heuristics, and compare the effectiveness of each search with $\chi^1$ and $\chi^2$, against the baseline of no pruning at all. Additionally, to test how well the pruning method works with a generally high-performant search, we also run the experiments using a greedy search algorithm and the FF (Fast-Forward) search heuristic (Hoffmann and Nebel 2001), which finds a plan to the delete relaxation of the problem from a state, and using the cost of that plan as a heuristic value for that state.

Comparing no pruning to $\chi^1$ when the $h_{max}$ search algorithm is used, and comparing no pruning to $\chi^2$ when the $h^2$ search algorithm is used, will help explore the theorem of exponential separation for $h_{max}$ and $h^2$ respectively, as they share the same heuristic, allowing us to directly see the effects of the early backtracking.

Figure 2 shows the results of each of these experiments for the *blocks* and *pipesworld-tankage* domains. These two domains were chosen in particular to emphasise the varying effectiveness of our approach, with *blocks* being somewhat of an outlier with its highly effective performance using certain parameters, and the more common case of *pipesworld-tankage* in which our approach generally underperforms. Note that we omit the case where the FF search heuristic is used with $\chi^2$, as only one problem was able to be solved in the specified time and memory limits.

## Pruning Using the $h_{max}$ Search Heuristic

In terms of states expanded, $\chi^2$ expands fewer states than $\chi^1$ most of the time, and both $\chi^2$ and $\chi^1$ *always* expand fewer states than no pruning. This shows that early backtracking is indeed occurring as expected. A particularly extreme case of this is *blocks* problem 18, where the difference of states between $\chi^2$ and no pruning is almost 24 million states, thus managing to prune 99.991% of states from the search. Despite this, $\chi^2$ is still approximately 25 seconds slower to find a solution, mainly due to the overhead of both generating the $H_T$ and $\chi_T$ clauses and the SAT solving at each search step.

Because of the significant amount of time generating $\chi_T$ in particular, $\chi^2$ tends to vastly underperform in terms of problems solved and solve time. A few exceptions exist, such as $\chi^2$ solving more problems than $\chi^1$ in the *blocks* domain, matching the number of problems that no pruning solved in that same domain. Also, there exists two problems in *blocks* (15 and 16) where $\chi^2$ is faster than $\chi^1$. These problems appear to be rare outlier cases for this particular domain, however, as $\chi^2$ is consistently the slowest option elsewhere.

Likewise, the last problem solved in both the *pipesworld-tankage* and *zenotravel* domains are solved faster using $\chi^1$ than no pruning. It would be tempting to believe that perhaps the time saved from using $\chi^1$ is beginning to appear

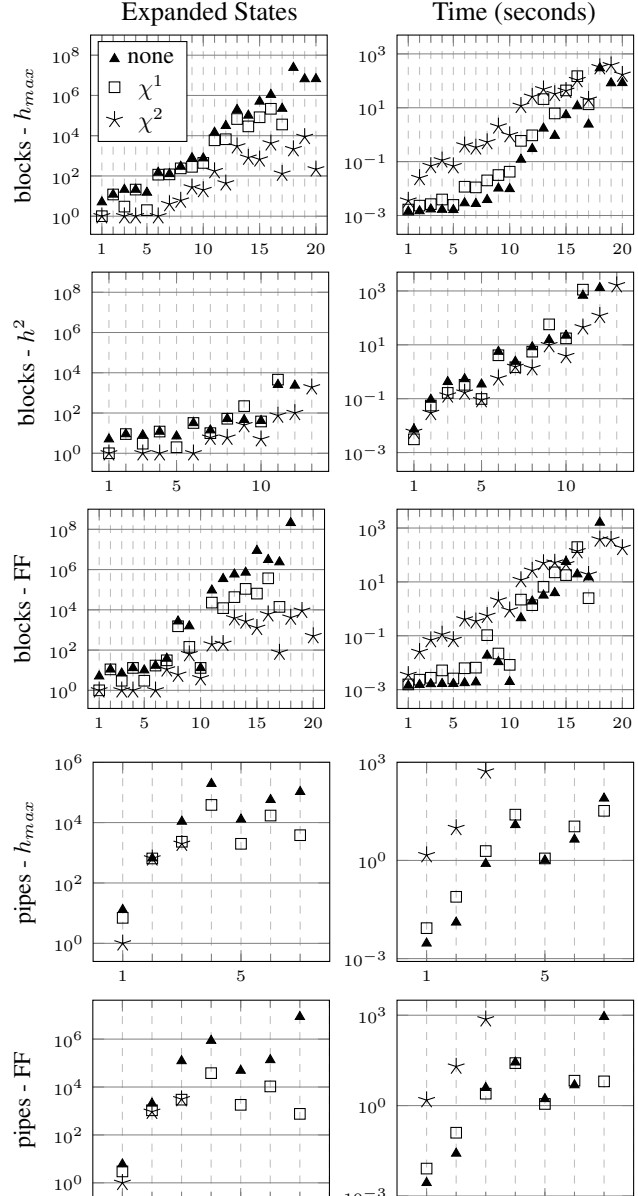

Figure 2: Various search heuristics (by row) and prune heuristics (by plot mark) for the *blocks* and *pipesworld-tankage* domains. Number of expanded states for each problem are displayed in the left column of plots, and total time to solve each problem are displayed in the right column. The $h^2$ search heuristic plots for the *pipesworld-tankage* domain have been omitted due to lack of data points. The x-axis represents the lexicographical order of the problems in the domain specified.

from that point onwards, and will only increase if more problems were to be solved. However, upon further inspection into these cases, they seem to also be coincidental outliers. Artificially giving the solver the optimal bound to search (to save time searching through unsolvable bounds) and solv-

ing further problems in these domains result in $\chi^1$ returning to underperformance immediately afterwards. This could be the result of UP managing to solve the problem entirely at some point, ending the search prematurely. In summary, it appears that heuristic pruning using $h_{max}$ as a search heuristic, while decreasing the search space to an often significant degree, is ineffective in terms of solve time.

## Pruning Using the $h^2$ Search Heuristic

For this search heuristic, $\chi^2$ expands fewer states than $\chi^1$ a majority of the time as well, though there are more exceptions in this case. Similarly, both $\chi^2$ and $\chi^1$ *almost* always expand fewer states than no pruning, but now there are a few problems where this is not the case. This shows that while pruning may decrease the total search space, it does not guarantee that a solution will be found earlier in the search.

The key difference in the $h^2$ search heuristic case, common among most domains, is that heuristic pruning is generally *faster* than using no pruning. This is particularly apparent in domains such as *miconic* and *philosophers*, where $\chi^1$ always results in the most efficient solve times. Overall, it appears that heuristic pruning using $h^2$ as a search heuristic is generally more effective than using no pruning, both in terms of expanded states and time.

For $h_{max}$, when comparing the results of no pruning and pruning with the same heuristic as the search, it seems that the exponential separation theorem from (Rintanen 2011) unfortunately does not occur in our hybrid approach, and doing so results in consistently slower solve times. On the other hand, the same comparison for the $h^2$ search heuristic shows that there could be an advantage to using heuristic pruning, and though the separation may not be exponential in practice, there is still a noteworthy performance gain.

## Pruning Using the FF Search Heuristic

With regards to states expanded in this case, $\chi^2$ generally expands significantly less states than no pruning. However, apart from the *blocks* domain, the difference between $\chi^2$ and $\chi^1$ is minimal. In some cases, $\chi^1$ even expands fewer states than $\chi^2$. The data overall appears to bear a resemblance to the data obtained from the $h_{max}$ search heuristic experiments, with the exception of the *miconic* domain, which tends to be a lot more chaotic and results in what appears to be a less stable difficulty progression.

In terms of solve time, $\chi^1$ manages to beat no pruning in a few more occasions, though it is still slower overall. $\chi^2$ remains consistently worse than any of the other options in terms of time efficiency. $\chi^1$ appears to shine the most in the *miconic* domain, where it beats the solve time of no pruning in 12 of the 50 problems solved. The solve time for this domain is quite variable in general when using the FF search heuristic, which is particularly clear in cases such as problems 43 and 45, which take 0.003 and 222 seconds respectively using FF, but 27 and 121 seconds using $h_{max}$. FF appears to be a lot less stable with solving problems such as these, which could allow for heuristic pruning to obtain an advantage by smoothing out cases where the heuristic

alone underperforms, guiding the search towards the solution faster than the overhead can overshadow.

The curious cases of *pipesworld-tankage* problem 7 and *zenotravel* problem 8 are even more pronounced here, with $\chi^1$ being even better than the previous experiments when it comes to beating no pruning with solve time. This suggests that more efficient search heuristics will only increase the advantages of heuristic pruning in problems where heuristic pruning has proven to be well-suited in terms of runtime.

All experiments considered, $\chi^2$ appears to be *especially* well-suited for the *blocks* domain, as it is able to solve more problems here than any other configuration. The states expanded for $\chi^2$ has a tendency to plateau while other configurations explode. *blocks* is quite a difficult domain to solve optimally in general, and it is possible that this advantageous performance could also occur in other planning domains that we have not experimented on.

In summary, while the speed of heuristic pruning is improved in some cases, using a performant heuristic such as FF still results in no heuristic pruning being faster in general.

## Conclusion and Future Work

Overall, our results showed that despite heuristic-based pruning resulting in fewer explored nodes, no pruning is the fastest method using either the $h_{max}$ or FF search heuristic, though $\chi^2$ is the fastest when using the $h^2$ search heuristic. Generally we find that the cost of generating the SAT representation of the heuristic—and regularly performing UP based inference on SAT problems—outweighs the benefit of the resulting reduced search space. This is a particularly big problem for $\chi^2$, where the generation of the SAT formula tends to dominate runtime.

A particularly interesting result is that $\chi^2$ performs very well in the *blocks* domain specifically, sometimes even beating no pruning in terms of time efficiency and problems solved. This shows us that there exists domains where heuristic-based pruning can be advantageous.

We find the following ideas to be compelling future research directions. Exploring the use of heuristic pruning where search is performed strategically using plan length bounds, such as decreasing from a known upper bound (Rintanen and Gretton 2013), query-based strategies (Streeter and Smith 2007), and non-sequential approaches (Rintanen 2004). Heuristic pruning works with any admissible heuristic, provided that it has a SAT implementation $\chi_T$. With that in mind, Pattern Database (PDB) heuristics (Culberson and Schaeffer 1998) are an interesting candidate for future exploration. Furthermore, with other types of search knowledge in mind, we expect pruning using knowledge other than the horizon bound, such as bounds one can derive using landmarks (Hoffmann, Porteous, and Sebastia 2004), to be a possible fruitful direction for future study. While our implementation uses the Direct Encoding to represent problems in SAT, there exist other encodings that could potentially be faster to generate, such as the SASE encoding (Huang, Chen, and Zhang 2010), ∃-Step encodings (Rintanen, Heljanko, and Niemelä 2006), or encodings that make use of a split representations (Robinson et al. 2009).

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
