# OpenReview forum: "A Study of the Power of Heuristic-based Pruning via SAT Planning"
_icaps-conference.org/ICAPS/2022/Workshop/HSDIP — HSDIP 2022_

### Official Review · Reviewer_fwwS · 2022-04-21
**Well written paper and relevant, however, missing a convincing motivation for the research**

**Confidence:** 4
**Overall Score:** Accept

**Review:**

In this paper, the nominated heuristic is implemented in SAT with the query formula encoding heuristic information. They start with presenting Definitions and Notations, then they move on to provide an implementation of the h^2 heuristic in SAT-based planning and how to use it as part of a search. Finally, they empirically evaluate SAT-based pruning in search by implementing heuristics h_max and h^2.

The topic of the paper is relevant to the workshop, clear planning paper, an interesting approach, and well-established related work. The part discussing the background is quite massive. In this case- it is a good fit. Here it is impotent to have such an extensive Definitions and Notations part in order to easily follow the rest of the paper.

The paper is well written, and the structure of the paper makes a lot of sense to me. The authors start with definitions, then discuss the implementation of heuristics in SAT and their encoding. Later they move on to explain how to use this heuristic as part of a search and finally conclude with an empirical evaluation. However, a part that I found missing is the motivation, especially due to the apologetic concern in the introduction (“Whether or not this method results in an overall advantage…”). Stating with such a doubt in your results should be accompanied with a strong motivation to explore this topic.

The technical results are nice and interesting, this is a contribution by itself, however, the contribution by the empirical evaluation is somewhat limited. The experiments are conducted on 24 planning domains but a plot is presented only for 2 domains – blocks and pipesworld. In addition, many of the results are integrated into the text and it was hard to follow what worked well and on which domain, I would suggest adding a table for a visual summary. It seems like the results are not providing a significant impact, which brings me to the motivation concern, to have low impact is something that might happen in research, but you should convince your audience that even though the contribution is limited the approach is interesting for itself and I couldn't find a convincing claim for that in your paper.

---

### Official Review · Reviewer_KiFJ · 2022-04-25
**New SAT encodings for heuristics**

**Confidence:** 4
**Overall Score:** Accept

**Review:**

The paper investigates how h1 and h2 heuristics can be encoded as a SAT instance and used for either (or both) of computing heuristic values or for pruning in a forward search planner. The work is very relevant to the HSDIP community, and I think this paper would be well received.

I would have liked to see a closer contrast to two existing works -- (1) Robinson's encoding for a relaxed suffix is referenced, but is quite close in spirit to what is being accomplished here; and (2) Zhang and Bacchus' paper, "MAXSAT Heuristics for Cost Optimal Planning" focuses on an encoding for h+. With respect to the latter, I would be curious about the author's motivation to use SAT-based technology rather than MaxSAT for the heuristic computation (perhaps the question only makes sense if full solutions are considered, and not just UP-restricted solutions).

One core question that struck me is why appeal to SAT at all when UP, from Algorithm 1, is the only component run. No decisions, smart clause learning (which could be carried from one instance to the next), etc, etc. If it was only unit propagation that was being used, how is it different from the fixed-point computations of h1 and h2 that we would typically see inside of a state-of-the-art planner? It feels as though the real strength would only come in the form of a proper SAT call. At the very least, you should be able to quantify how often unit prop fails to prove SAT or UNSAT -vs- those cases where the UP-only strategy works.

Another question that came to mind was the encoding around Type 1. First, it seems as though you should be able to describe the theory without extra variables -- have notation that represents a term (conjunction) capturing the set of variables, and then use that throughout the formulae (allowing the Tseitin encoding to create whatever variables you need). If the resulting theory is simple enough, then you may save needing these variables at all. Also, it seems as though you would want an if-and-only-if for Type 1 (only one direction of the 2-way implication is encoded).

Type 2 & 3 were a bit hard to follow, so any example that would illustrate things would be quite helpful for them.

While looking at the overall approach, especially if you were to extend to the full SAT/UNSAT case (and not just what UP can provide), it seems as though giving knowledge compilation a try might be worthwhile. For example, using a compiler such as d4 for each time bound would let you compile the full space of theories so that heuristic calculation / pruning can be handled with a lookup on the resulting d-DNNF.

Another potential change that comes to mind is when things are evaluated. It seems as though nodes are evaluated as they are generated, but lazy evaluation (when expanded) is a very common approach in modern planners that may be useful here.

Finally, it seems as though there is a missed opportunity with the setting you've proposed. Given that the heuristic computation is done _via SAT_, any approach that creates additional constraints can be included. Overlayed LTL, axioms (perhaps?), conditional effects, etc, etc. If it can be modelled logically, then you have a natural means of extending to it. This makes the contribution powerful, but that needs to be reflected in the text.

All in all, this is interesting work, and I look forward to seeing it presented at the workshop!

---

### Public Comment · (anonymous) · 2022-04-28
**Response to Reviewers**

Thank you to both reviewers for your constructive and valuable feedback on our submission. We will be sure to take into account these suggestions to present a polished work for the camera-ready version. Regarding a few of the explicit questions put forward, here are our responses:

Reviewer KiFj:

1. Unlike Robinson et al. we do not use a SAT-based inference procedure as the basis for reasoning about a heuristic numerically. Rather, we augment the representation of the SAT problem so that UP triggers backtracking in the context of a nominal SAT-based planning scheme. It happens that the modified encoding is informed by a known heuristic, and that some of the modified SAT-based search behaviour can be characterised in terms of numeric heuristic information and the fixed planning horizon we have in IDA*. So, unlike Robinson et al., here we never explicitly reason about numbers, or numeric optimisation in SAT. Note also, we only study unit cost planning problem instances.

2. Following Rintanen and Huang, we have only considered UP inference. Using something more powerful, such as a sound and complete SAT procedure, perhaps with a decision budget, is a great idea. Our investigation is focused on whether or not the promise of Rintanen's conception plays out in the benchmarks. So, drifting from UP to more powerful inference would not be in the spirit of our project and original conception of encoding heuristics in SAT by Rintanen.

3. There are indeed many more possible representations than we have described here. We followed Huang more-or-less exactly.

4. Your suggestion about d4 does sound extremely interesting, and is definitely worth future investigation.

5. We were being faithful to the Theorems in the Rintanen paper by not considering lazy evaluations and other potential search optimisations. If time performance was the focus of our study, it would indeed likely be the right choice to consider synergies with other search optimisations. We wanted to understand if the search space size impact, in practice, resembled what Rintanen had indicated in his headline theorem.

6. There are indeed many ideas for expanding our study, such as you have suggested (LTL, etc.), that are missed. Many thanks for suggesting those interesting future research directions.

Reviewer fwwS:

1. Regarding impact: Our contribution is an empirical study following the publications of Rintanen and Huang, of novel representations of the planning problem in SAT. We are focused on novel empirical measurements, and are the first to empirically study those results.

2. It is true that a significant amount of the empirical results were not able to be presented visually. We plan to improve the clarity of the results for the empirical evaluation so that it is easier to follow and conclude results from.

We thank you again for the time and effort in reviewing this submission, and the useful suggestions put forward.

---

> ### Comment · Reviewer_KiFJ · 2022-05-02
> **Emphasize the aim**
>
> > We wanted to understand if the search space size impact, in practice, resembled what Rintanen had indicated in his headline theorem.
>
> This didn't quite jump out at me while reading the paper. I'd suggest highlighting it early on in the paper (abstract and/or intro) as a driving force behind the design decisions used in the work.
>
> Thanks for the detailed response!